# Learning from Explanations with Neural Execution Tree

**Ziqi Wang**[1][*] **Yujia Qin**[1][*], **Wenxuan Zhou**[2]**, Jun Yan**[2]**, Qinyuan Ye**[2]**, Leonardo Neves**[3]**,
Zhiyuan Liu**[1]**, Xiang Ren**[2]

Tsinghua University[1], University of Southern California[2], Snap Research[3]

`{ziqi-wan16, qinyj16}@mails.tsinghua.edu.cn, {liuzy}@tsinghua.edu.cn,`
`{zhouwenx, yanjun, qinyuany, xiangren}@usc.edu, {lneves}@snap.com`

## Abstract

While deep neural networks have achieved impressive performance on a range of NLP tasks, these data-hungry models heavily rely on labeled data, which restricts their applications in scenarios where data annotation is expensive. Natural language (NL) explanations have been demonstrated very useful additional supervision, which can provide sufficient domain knowledge for generating more labeled data over new instances, while the annotation time only doubles. However, directly applying them for augmenting model learning encounters two challenges: (1) NL explanations are unstructured and inherently compositional, which asks for a modularized model to represent their semantics, (2) NL explanations often have large numbers of linguistic variants, resulting in low recall and limited generalization ability. In this paper, we propose a novel **N**eural **Ex**ecution **T**ree (NExT) framework[1] to augment training data for text classification using NL explanations. After transforming NL explanations into executable logical forms by semantic parsing, NExT generalizes different types of actions specified by the logical forms for labeling data instances, which substantially increases the coverage of each NL explanation. Experiments on two NLP tasks (relation extraction and sentiment analysis) demonstrate its superiority over baseline methods. Its extension to multi-hop question answering achieves performance gain with light annotation effort.

## 1 Introduction

Deep neural networks have achieved state-of-the-art performance on a wide range of natural language processing tasks. However, they usually require massive labeled data, which restricts their applications in scenarios where data annotation is expensive. The traditional way of providing supervision is human-generated labels. See Figure 1 as an example. The sentiment polarity of the sentence *"Quality ingredients preparation all around, and a very fair price for NYC"* can be labeled as *"Positive"*. However, the label itself does not provide information about how the decision is made. A more informative method is to allow annotators to explain their decisions in natural language so that the annotation can be generalized to other examples. Such an explanation can be *"Positive, because the word price is directly preceded by fair"*, which can be generalized to other instances like *"It has delicious food with a fair price"*. Natural language (NL) explanations have shown effectiveness in providing additional supervision, especially in low-resource settings (Srivastava et al., 2017; Hancock et al., 2018). Also, they can be easily collected from human annotators without significantly increasing annotation efforts.

However, exploiting NL explanations as supervision is challenging due to the complex nature of human languages. First of all, textual data is not well-structured, and thus we have to parse explanations into logical forms for machine to better utilize them. Also, linguistic variants are ubiquitous, which makes it difficult to generalize an NL explanation for matching sentences that are semantically

---

[*]   Equal contribution. The order is decided by a coin toss. The work was done when visiting USC.
[1]   Project: `http://inklab.usc.edu/project-NExT/`
      Code: `https://github.com/INK-USC/NExT`

equivalent but having different word usage. When we perform exact matching with the previous example explanation, it fails to annotate sentences with "*reasonable price*" or "*good deal*".

Attempts have been made to train classifiers with NL explanations. Srivastava et al. (2017) use NL explanations as additional features of data. They map explanations to logical forms with a semantic parser and use them to generate binary features for all instances. Hancock et al. (2018) employ a rule-based semantic parser to get logical forms (i.e. "labeling function") from NL explanations that generate noisy labeled datasets used for training models. While both methods claim huge performance improvements, they neglect the importance of linguistic variants, thus resulting in a very low recall. Also, their methods of evaluating explanations on new instances are oversimplified (e.g. comparison/logic operators), making their methods overly confident. In the above example, sentence *"Decent sushi at a fair enough price"* will be rejected because of the *"directly preceded"* requirement.

| |
|---|
| Sentence: quality ingredients preparation all around, and a very fair price for NYC. |
| What is the sentiment polarity w.r.t. "price"? |
| Label: Positive |
| Explanation: because the word "price" is directly preceded by fair. |

matching from corpus

| |
|---|
| Sentence: it has delicious food with a fair price. |

Figure 1: Matching new instances from raw corpus using natural language explanations.

To address these issues, we propose **Neural Execution Tree** (NExT) framework for deep neural networks to learn from NL explanations, as illustrated in Figure 2. Given a raw corpus and a set of NL explanations, we first parse the NL explanations into machine-actionable logical forms by a combinatory categorial grammar (CCG) based semantic parser. Different from previous work, we "soften" the annotation process by generalizing the predicates using neural module networks and changing the labeling process from exact matching to fuzzy matching. We introduce four types of matching modules in total, namely *String Matching Module*, *Soft Counting Module*, *Logical Calculation Module*, and *Deterministic Function Module*. We calculate the matching scores and find for each instance the most similar logical form. Thus, all instances in the raw corpus can be assigned a label and used to train neural models.

The major **contributions** of our work are summarized as follows: (1) We propose a novel NExT framework to utilize NL explanations. NExT is able to model the compositionality of NL explanations and improve the generalization ability of NL explanations so that neural models can leverage unlabeled data for augmenting model training. (2) We conduct extensive experiments on two representative tasks (relation extraction and sentiment analysis). Experimental results demonstrate the superiority of NExT over various baselines. Also, we adapted NExT for multi-hop question answering task, in which it achieves performance improvement with only 21 explanations and 5 rules.

## 2 LEARNING TO AUGMENT SEQUENCE MODELS WITH NL EXPLANATIONS

This section first talks about basic concepts and notations for our problem definition. Then we give a brief overview of our approach, followed by details of each stage.

**Problem Definition.** We consider the task of training classifiers with natural language explanations for text classification (e.g., relation extraction and sentiment analysis) in a low-resource setting. Specifically, given a raw corpus $\mathcal{S} = \{\mathbf{x}_i\}_{i=1}^N \subseteq \mathcal{X}$ and a predefined label set $\mathcal{Y}$, our goal is to learn a classifier $f_c : \mathcal{X} \to \mathcal{Y}$. We ask human annotators to view a subset $\mathcal{S}'$ of the corpus $\mathcal{S}$ and provide for each instance $\mathbf{x} \in \mathcal{S}'$ a label $\mathbf{y}$ and an explanation $\mathbf{e}$, which explains why $\mathbf{x}$ should receive $\mathbf{y}$. Note that $|\mathcal{S}'| \ll |\mathcal{S}|$, which requires our framework to learn with very limited human supervision.

**Approach Overview.** We develop a multi-stage learning framework to leverage NL explanations in a weakly-supervised setting. An overview of our framework is depicted in Fig. 2. Our NExT framework consists of three stages, namely explanation parsing stage, dataset partition stage, and joint model learning stage.

**Explanation Parsing.** To leverage the unstructured human explanations $\mathcal{E} = \{\mathbf{e}_j\}_{j=1}^{|\mathcal{S}'|}$, we turn them into machine-actionable logical forms (i.e., labeling functions) (Ratner et al., 2016), which can be denoted as $\mathcal{F} = \{f_j : \mathcal{X} \to \{0, 1\}\}_{j=1}^{|\mathcal{S}'|}$, where 1 indicates the logical form matches the

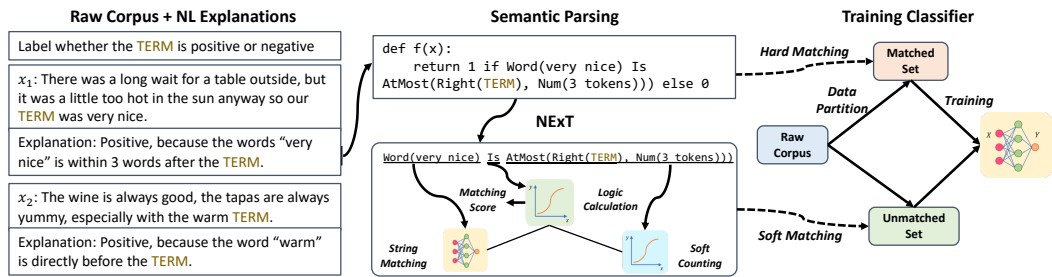

Figure 2: Overview of the NExT Framework. Natural language explanations are firstly parsed into logical forms. Then we partition the raw corpus $\mathcal{S}$ into labeled dataset $\mathcal{S}_a$ and unlabeled dataset $\mathcal{S}_u = \mathcal{S} - \{\mathbf{x}_i^a\}_{i=1}^{N_a}$. We use *matching modules* to provide supervision on $\mathcal{S}_u$. Finally, supervision from both $\mathcal{S}_a$ and $\mathcal{S}_u$ is fed into a classifier.

input sequence and 0 otherwise. To access the labels, we introduce a function $h : \mathcal{F} \to \mathcal{Y}$ that maps each logical form $f_j$ to the label $y_j$ of its explanation $\mathbf{e}_j$. Examples are given in Fig. 2. We use Combinatory Categorial Grammar (CCG) based semantic parsing (Zettlemoyer & Collins, 2012; Artzi et al., 2015), an approach that couples syntax with semantics, to convert each NL explanation $\mathbf{e}_j$ to a logical form $f_j$.

Following Srivastava et al. (2017), we first compile a domain lexicon that maps each word to its syntax and logical predicate. Frequently-used predicates are listed in the Appendix. For each explanation, the parser can generate many possible logical forms based on CCG grammar. To identify the correct one from these logical forms, we use a feature vector $\phi(f) \in \mathcal{R}^d$ with each element counting the number of applications of a particular CCG combinator (similar to Zettlemoyer & Collins (2007)). Specifically, given an explanation $\mathbf{e}_i$, the semantic parser parameterized by $\boldsymbol{\theta} \in \mathcal{R}^d$ outputs a probability distribution over all possible logical forms $\mathscr{Z}_{\mathbf{e}_i}$. The probability of a feasible logical form can be calculated as:

$$P_\theta(f|\mathbf{e}_i) = \frac{\exp \boldsymbol{\theta}^T \boldsymbol{\phi}(f)}{\sum_{f':f'\in\mathcal{Z}_{\mathbf{e}_i}} \exp \boldsymbol{\theta}^T \boldsymbol{\phi}(f')}.$$

To learn $\theta$, we maximize the probability of $\mathbf{y}_i$ given $\mathbf{e}_i$ calculated by marginalizing over all logical forms that match $\mathbf{x}_i$ (similar to Liang et al. (2013)). Formally, the objective function is defined as:

$$L_{parser} = \sum_{i=1}^{|\mathcal{S}'|} \log \big( \sum_{f:f(\mathbf{x}_i)=1 \,\wedge\, h(f)=y_i} P_\theta(f|\mathbf{e}_i) \big).$$

When the optimal $\theta^*$ is derived using gradient-based method, the parsing result for $\mathbf{e}_i$ is defined as $f_i = \arg\max_f P_{\theta^*}(f|\mathbf{e}_i)$.

**Dataset Partition.** After we parse explanations $\{\mathbf{e}_i\}_{i=1}^{|\mathcal{S}'|}$ into $\mathcal{F} = \{f_i\}_{i=1}^{|\mathcal{S}'|}$, where each $f_i$ corresponds to $\mathbf{e}_i$, we use $\mathcal{F}$ to find exact matches in $\mathcal{S}$ and pair them with the corresponding labels. We denote the number of instances getting labeled by exact matching as $N_a$. As a result, $\mathcal{S}$ is partitioned into a labeled dataset $\mathcal{S}_a = \{(\mathbf{x}_i^a, y_i^a)\}_{i=1}^{N_a}$, and an unlabeled dataset $\mathcal{S}_u = \mathcal{S} - \{\mathbf{x}_i^a\}_{i=1}^{N_a} = \{\mathbf{x}_j^u\}_{j=1}^{N_u}$ where $N_u = |\mathcal{S}| - N_a$.

**Joint Model Learning.** The exactly matched $\mathcal{S}_a$ can be directly used to train a classifier while informative instances in $\mathcal{S}_u$ are left untouched. We propose several neural module networks, which relax constraints in each logic form $f_j$ and substantially improve the rule coverage in $\mathcal{S}_u$. Classifiers will benefit from these soft-matched and pseudo-labeled instances. Trainable parameters in neural module networks are jointly optimized with the classifier. Details of each module and joint training method will be introduced in the next section.

# 3 NEURAL EXECUTION TREE

Given a logical form $f$ and a sentence $\mathbf{x}$, NExT will output a matching score $u_s \in [0,1]$, which indicates how likely the sentence $\mathbf{x}$ satisfies the logical form $f$ and thus should be given the corresponding label $h(f)$. Specifically, NExT comprises of four modules to deal with four categories of

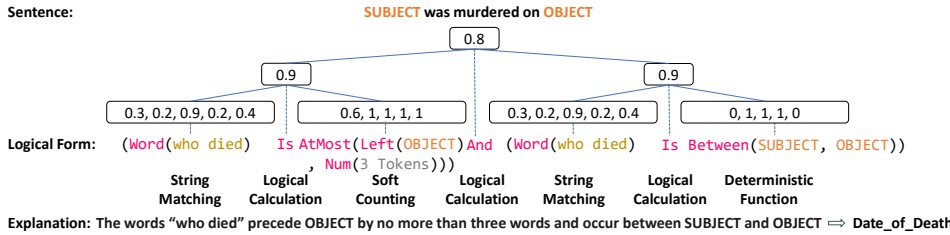

Figure 3: Neural Execution Tree (NExT) softly executes the logical form on the sentence.

predicates, namely *String Matching Module*, *Soft Counting Module*, *Deterministic Function Module*, and *Logical Calculation Module*. Any complex logical form can be disassembled into clauses containing these four categories of predicates. The four modules are then used to evaluate each clause and then the whole logical form in a softened way. Fig. 3 shows how NExT builds the execution tree from an NL explanation and how it evaluates an unlabeled sentence. We show in the figure the corresponding module for each predicate in the logical form.

## 3.1 MODULES IN NEXT

**String Matching Module.** Given a keyword query $\mathbf{q}$ derived from an explanation and an input sequence $\mathbf{x} = [w_1, w_2, ..., w_n]$, the string matching module $f_s(\mathbf{x}, \mathbf{q})$ returns a sequence of scores $[s_1, s_2, ..., s_n]$ indicating the similarity between each token $w_i$ and the query $\mathbf{q}$. Previous work implements this operation by exact keyword searching, while we augment the module with neural networks to enable capturing semantically similar words. Inspired by Li et al. (2018), for token $w_i$, we first generate $N_c$ contexts by sliding windows of various lengths. For example, if the maximum window size is 2, the contexts $\mathbf{c}_{i0}, \mathbf{c}_{i1}, \mathbf{c}_{i2}$ of token $w_i$ are $[w_i]$, $[w_{i-1}; w_i]$ and $[w_i; w_{i+1}]$ respectively. Then we encode each context $\mathbf{c}_{ij}$ to a vector $\mathbf{z}_{\mathbf{c}_{ij}}$ by feeding pre-trained word embeddings into a bi-directional LSTM encoder (Hochreiter & Schmidhuber, 1997). All hidden layers of BiLSTM are then summarized by an attention layer (Bahdanau et al., 2014). For keyword query $\mathbf{q}$, we directly encode it into vector $\mathbf{z}_q$ by bi-LSTM and attention. Finally, scores of sentence $\mathbf{x}$ and query $\mathbf{q}$ are calculated by aggregating similarity scores from different sliding windows:

$$\boldsymbol{M}_{ij}(\mathbf{x}, \mathbf{q}) = \cos(\mathbf{z}_{\mathbf{c}_{ij}} \boldsymbol{D}, \mathbf{z}_{\mathbf{q}} \boldsymbol{D}), \quad f_s(\mathbf{x}, \mathbf{q}) = \boldsymbol{M}(\mathbf{x}, \mathbf{q}) \mathbf{v},$$

where $\boldsymbol{D}$ is a trainable diagonal matrix, $\mathbf{v} \in \mathcal{R}^{N_c}$ is the trainable weight of each sliding window.

Parameters in the string matching module need to be learned with data in the form of *(sentence, keyword, label)*. To build a training set for learning string matching, we randomly select spans of consecutive words as keyword queries in the training data. Each query is paired with the sentence it comes from. The synthesized dataset is denoted as $\{\mathbf{x}_i, \mathbf{q}_i, \mathbf{k}_i\}_{i=1}^{N_{syn}}$, where $\mathbf{k}_{ij}$ will take the value of 1 if $\mathbf{q}$ is extracted from $\mathbf{x}_{ij}$ and 0 otherwise. The loss function is defined as the binary cross-entropy loss, as follows.

$$L_{find} = -\frac{1}{N_{syn}} \sum_{i=1}^{N_{syn}} \frac{1}{|\mathbf{k}_i|} \cdot (\mathbf{k}_i \log f_s(\mathbf{x}_i, \mathbf{q}_i) + (1 - \mathbf{k}_i) \log(1 - f_s(\mathbf{x}_i, \mathbf{q}_i))).$$

While pretraining with $L_{find}$ enables matching similar words, this unsupervised distributional method is poor at learning their semantic meanings. For example, the word "good" will have relatively low similarity to "great" because there is no such training data. To solve this problem, we borrow the idea of word retrofitting (Faruqui et al., 2014) and adopt a contrastive loss (Neculoiu et al., 2016) to incorporate semantic knowledge in training. We use the keyword queries in labeling functions as supervision. Intuitively, the semantic meaning of two queries should be similar if they appear in the same class of labeling functions and dissimilar otherwise. More specifically, for a query $\mathbf{q}$, we denote queries in the same class of labeling functions as $\mathcal{Q}_+$ and queries in different classes of labeling functions as $\mathcal{Q}_-$. The similarity loss is defined as:

$$L_{sim} = \max_{\mathbf{q}_1 \in \mathcal{Q}_+} \{(\tau - \cos(\mathbf{z}_{\mathbf{q}} \boldsymbol{D}, \mathbf{z}_{\mathbf{q}_1} \boldsymbol{D}))_+^2\} + \max_{\mathbf{q}_2 \in \mathcal{Q}_-} \{\cos(\mathbf{z}_{\mathbf{q}} \boldsymbol{D}, \mathbf{z}_{\mathbf{q}_2} \boldsymbol{D})_+^2\}.$$

The overall objective function for string matching module is:

$$L_{string} = L_{find} + \gamma \cdot L_{sim}, \tag{1}$$

---

**Algorithm 1:** Learning on Unlabeled Data with NExT

---

**Input:** Labeled data $\mathcal{S}_a = \{(\mathbf{x}_i^a, y_i^a)\}_{i=1}^{N_a}$, unlabeled data $\mathcal{S}_u = \{\mathbf{x}_j^u\}_{j=1}^{N_u}$, and logical forms $\mathcal{F} = \{f_k\}_{k=1}^{|\mathcal{S}'|}$.
**Output:** A classifier $f_c : \mathcal{X} \rightarrow \mathcal{Y}$.
Pretrain *String Matching Module* in NExT *w.r.t.* $L_{string}$ using Eq. 1.
**while** *not converge* **do**

    Sample a labeled batch $\mathcal{B}_a = \{(\mathbf{x}_i^a, y_i^a)\}_{i=1}^n$ from $S_a$, and an unlabeled batch $\mathcal{B}_u = \{\mathbf{x}_j^u\}_{j=1}^m$ from $\mathcal{S}_u$.
    **foreach** $\mathbf{x}_j^u \in \mathcal{B}_u$ **do**
       |  Calculate a pseudo label $y_j^u$ for $\mathbf{x}_j^u$ with confidence $u_j$ using NExT and $\mathcal{F}$.
    Normalize matching scores $\{u_j\}_{j=1}^m$ to get $\{\omega_j\}_{j=1}^m$ based on Eq. 3.
    Calculate $L_a$ using Eq. 2, $L_u$ using Eq. 4, $L$ using Eq. 5.
    Update $f_c$ and *String Matching Module* in NExT *w.r.t.* $L_{total}$.

---

where $\gamma$ is a hyper-parameter. We pretrain the string matching module for better initialization.

**Soft Counting Module.** The soft counting module aims to relax the counting (distance) constraints defined by NL explanations. For a counting constraint *precede object by no more than three words*, the soft counting module outputs a matching score indicating to which extent an anchor word (TERM, SUBJECT, and OBJECT) satisfies the constraint. The score is set to 1 if the position of the anchor word strictly satisfies the constraint, and will decrease if the constraint is broken. For simplicity, we allow an additional range in which the score is set to $\mu \in (0, 1)$, which is a hyper-parameter controlling the constraints.

**Deterministic Function Module.** The deterministic function module deals with deterministic predicates like "Left" and "Right", which can only be exactly matched because a string is either right or left of an anchor word. Therefore, the probability it outputs should be either 0 or 1. The Deterministic Function Module deals with all these predicates and outputs a mask sequence, which is fed into the tree structure and combined with other information.

**Logical Calculation Module.** The logical calculation module acts as a score aggregator. It can aggregate scores given by: (1) a string matching module and a soft counting module/deterministic function module (triggered by predicates such as "Is" and "Occur") (2) two clauses that have been evaluated with a score respectively (triggered by predicates such as "And" and "Or").

In the first case, the logical calculation module will calculate the element-wise products of the score sequence provided by the string matching module and the mask sequence provided by the soft counting module/deterministic function module. It then uses max pooling to calculate the matching score of the current clause. In the second case, the logical calculation module will aggregate the scores of at least one clause based on the logic operation. The rules are defined as follows.

$$p_1 \wedge p_2 = \max(p_1 + p_2 - 1, 0), \quad p_1 \vee p_2 = \min(p_1 + p_2, 1), \quad \neg p = 1 - p,$$

where $p$ is the score of the input clause.

### 3.2 Augmenting Model Learning with NExT

As described in Algo. 1, in each iteration, we sample two batches $\mathcal{B}_a$ and $\mathcal{B}_u$ from $\mathcal{S}_a$ and $\mathcal{S}_u$. We conduct supervised learning on $\mathcal{B}_a$. The labeled loss function is calculated as:

$$L_a = -\frac{1}{N_a} \sum_{(\mathbf{x}_i^a, y_i^a) \in \mathcal{B}_a} \log p(y_i^a | \mathbf{x}_i^a). \tag{2}$$

To leverage $\mathcal{B}_u$, which is also informative, for each instance $\mathbf{x}_j^u \in \mathcal{B}_u$, we can use our *matching modules* to compute its matching score with every logical form. The most probable logical form that matched with $\mathbf{x}_j^u$ is denoted as $\mathbf{y}_j^u$ [2], along with the matching score $u_j$. To ensure the scale of the unlabeled loss is comparable to labeled loss, we normalize the matching scores among pseudo-labeled instances in $\mathcal{B}_u$ as:

$$\omega_j = \frac{\exp(\theta_t u_j)}{\sum_{k=1}^{|\mathcal{B}_u|} \exp(\theta_t u_k)}, \tag{3}$$

---

[2] *None* label (e.g. *No_Relation* for relation extraction and *Neutral* for sentiment analysis) usually lacks explanations. If the entropy of downstream model prediction distribution over labels is lower than a threshold, a *None* label will be given.

| Dataset | exps | categs | avg ops | logic/% | assertion/% | position/% | counting/% | acc/% |
|---------|------|--------|---------|---------|-------------|------------|------------|-------|
| TACRED | 170 | 13 | 8.2 | 25.8 | 21.3 | 21.4 | 12.4 | 95.3 |
| SemEval | 203 | 9 | 4.2 | 32.7 | 15.9 | 26.3 | 5.5 | 84.2 |
| Laptop | 40 | 8 | 3.9 | 0.0 | 23.8 | 23.8 | 17.5 | 87.2 |
| Restaurant | 45 | 9 | 9.6 | 2.8 | 25.4 | 26.1 | 16.2 | 88.2 |

Table 1: **Statistics for Human-curated Explanations and Evaluation of Semantic Parsing.** We report the number of NL explanations (*exps*), categories of predicates (*categs*) and operator compositions per explanation (*avg ops*) respectively. We also report the proportions of different types of predicates, where *logic* denotes logical operators (*And, Or*), *assertion* denotes assertion predicates (*Occur, Contains*), *position* denotes position predicates (*Right, Between*) and *counting* denotes counting predicates (*MoreThan, AtMost*). We summarize the accuracy (acc) of semantic parsing based on human evaluation.

where $k$ is the index of the instance and hyperparameter $\theta_t$ (temperature) controls the shape of normalized scores' distribution. Based on that, the unlabeled loss is calculated as:

$$L_u = - \sum_{(\mathbf{x}_j^u \in \mathcal{B}_u)} \omega_j \log p(y_j^u | \mathbf{x}_j^u). \tag{4}$$

Note that the string matching module is also trainable and plays a vital role in NExT. We jointly train it with the classifier by optimizing:

$$L_{total} = L_a + \alpha \cdot L_u + \beta \cdot L_{string}, \tag{5}$$

where $\alpha$ and $\beta$ are hyper-parameters.

## 4 EXPERIMENTS

**Tasks and Datasets.** We conduct experiments on two tasks: relation extraction and aspect-term-level sentiment analysis. Relation extraction (RE) aims to identify the relation type between two entities in a sentence. For example, given a sentence *Steve Jobs founded Apple Inc*, we want to extract a triple (*Steve Jobs*, *Apple Inc.*, *Founder*). For RE we choose two datasets, TACRED (Zhang et al., 2017) and SemEval (Hendrickx et al., 2009) in our experiments. Aspect-term-level sentiment analysis (SA) aims to decide the sentiment polarity with regard to the given aspect term. For example, given a sentence *Quality ingredients preparation all around, and a very fair price for NYC*, the sentiment polarity of the aspect term *price* is positive, the explanation can be *The word price is directly preceded by fair*. For this task we use two customer review datasets, Restaurant and Laptop, which are part of SemEval 2014 Task 4.

**Explanation Collection.** We use Amazon Mechanical Turk to collect explanations for a randomly sampled set of instances in each dataset. Turkers are prompted with a list of selected predicates (see Appendix) and several examples of NL explanations. Examples of collected explanations are listed in Appendix. Statistics of curated explanations and intrinsic evaluation results of semantic parsing are summarized in Table 1. To ensure a low-resource setting (i.e., $|\mathcal{S}'| \ll |\mathcal{S}|$), in each experiment we only use a random subset of collected explanations.

**Compared Methods.** As mentioned in Sec. 2, logical forms partition unlabeled corpus $\mathcal{S}$ into labeled set $\mathcal{S}_a$ and unlabeled set $\mathcal{S}_u$. Labeled set $\mathcal{S}_a$ can be directly utilized by supervised learning methods. (1) **CBOW-GloVe** uses bag-of-words (Mikolov et al., 2013) on GloVe embeddings (Pennington et al., 2014) to represent an instance, or surface patterns in NL explanations. It then annotates the sentence with the label of its most similar surface pattern (by cosine similarity). (2) **PCNN** (Zeng et al., 2015) uses piece-wise max-pooling to aggregate CNN-generated features. (3) **LSTM+ATT** (Bahdanau et al., 2014) adds an attention layer onto LSTM to encode a sequence. (4) **PA-LSTM** (Zhang et al., 2017) combines LSTM with an entity-position aware attention to conduct relation extraction. (5) **ATAE-LSTM** (Wang et al., 2016) combines the aspect term information into both embedding layer and attention layer to help the model concentrate on different parts of a sentence.

For semi-supervised baselines, unlabeled data $\mathcal{S}_u$ is also introduced for training. For methods requiring rules as input, we use surface pattern-based rules transferred from explanations. Compared

|  | TACRED | SemEval |
|---|---|---|
| LF ($\mathcal{E}$) | 23.33 | 33.86 |
| CBOW-GloVe ($\mathcal{R} + \mathcal{S}$) | 34.6±0.4 | 48.8±1.1 |
| PCNN ($\mathcal{S}_a$) | 34.8±0.9 | 41.8±1.2 |
| PA-LSTM ($\mathcal{S}_a$) | 41.3±0.8 | 57.3±1.5 |
| BiLSTM+ATT ($\mathcal{S}_a$) | 41.4±1.0 | 58.0±1.6 |
| BiLSTM+ATT ($\mathcal{S}_l$) | 30.4±1.4 | 54.1±1.0 |
| Self Training ($\mathcal{S}_a + \mathcal{S}_u$) | 41.7±1.5 | 55.2±0.8 |
| Pseudo Labeling ($\mathcal{S}_a + \mathcal{S}_u$) | 41.5±1.2 | 53.5±1.2 |
| Mean Teacher ($\mathcal{S}_a + \mathcal{S}_u$) | 40.8±0.9 | 56.0±1.1 |
| Mean Teacher ($\mathcal{S}_l + \mathcal{S}_{lu}$) | 25.9±2.2 | 52.2±0.7 |
| DualRE ($\mathcal{S}_a + \mathcal{S}_u$) | 32.6±0.7 | 61.7±0.9 |
| Data Programming ($\mathcal{E} + \mathcal{S}$) | 30.8±2.4 | 43.9±2.4 |
| NExT ($\mathcal{E} + \mathcal{S}$) | **45.6±0.4** | **63.5±1.0** |

(a) Relation Extraction

|  | Restaurant | Laptop |
|---|---|---|
| LF ($\mathcal{E}$) | 7.7 | 13.1 |
| CBOW-GloVe ($\mathcal{R} + \mathcal{S}$) | 68.5±2.9 | 61.5±1.3 |
| PCNN ($\mathcal{S}_a$) | 72.6±1.2 | 60.9±1.1 |
| ATAE-LSTM ($\mathcal{S}_a$) | 71.1±0.4 | 56.2±3.6 |
| ATAE-LSTM ($\mathcal{S}_l$) | 71.4±0.5 | 52.0±1.4 |
| Self Training ($\mathcal{S}_a + \mathcal{S}_u$) | 71.2±0.5 | 57.6±2.1 |
| Pseudo Labeling ($\mathcal{S}_a + S_u$) | 70.9±0.4 | 58.0±1.9 |
| Mean Teacher ($\mathcal{S}_a + \mathcal{S}_u$) | 72.0±1.5 | 62.1±2.3 |
| Mean Teacher ($\mathcal{S}_l + \mathcal{S}_{lu}$) | 74.1±0.4 | 61.7±3.7 |
| Data Programming ($\mathcal{E} + \mathcal{S}$) | 71.2±0.0 | 61.5±0.1 |
| NExT ($\mathcal{E} + \mathcal{S}$) | **75.8±0.8** | **62.8±1.9** |

(b) Sentiment Analysis

Table 2: **Experiment results on Relation Extraction and Sentiment Analysis.** Average and standard deviation of F1 scores (%) over multiple runs are reported (5 runs for RE and 10 runs for SA). $LF(\mathcal{E})$ denotes directly applying logical forms onto explanations. Bracket behind each method illustrates corresponding data used in the method. $\mathcal{S}$ denotes training data without labels, $\mathcal{E}$ denotes explanations, $\mathcal{R}$ denotes surface pattern rules transformed from explanations; $\mathcal{S}_a$ denotes labeled data annotated with explanations, $\mathcal{S}_u$ denotes the remaining unlabeled data. $\mathcal{S}_l$ denotes labeled data annotated using same time as creating explanations $\mathcal{E}$, $\mathcal{S}_{lu}$ denotes remaining unlabeled data corresponding to $\mathcal{S}_l$.

semi-supervised methods include: (1) **Pseudo-Labeling** (Lee, 2013) first trains a classifier on labeled dataset, then generates pseudo labels for unlabeled data using the classifier by selecting the labels with maximum predicted probability. (2) **Self-Training** (Rosenberg et al., 2005) proposes to expand the labeled data by selecting a batch of unlabeled data that has the highest confidence and generate pseudo-labels for them. The method stops until the unlabeled data is used up. (3) **Mean-Teacher** (Tarvainen & Valpola, 2017) averages model weights instead of label predictions and assumes similar data points should have similar outputs. (4) **DualRE** (Lin et al., 2019) jointly trains a relation prediction module and a retrieval module.

Learning from explanations is categorized as a third setting. Explanation-guided pseudo labels are generated for a downstream classifier. (1) **Data Programming** (Hancock et al., 2018; Ratner et al., 2016) aggregates results of strict labeling functions for each instance and uses these pseudo-labels to train a classifier. (2) **NExT** (proposed work) softly applies logical forms to get annotations for unlabeled instances and trains a downstream classifier with these pseudo-labeled instances. The downstream classifier is BiLSTM+ATT for relation extraction and ATAE-LSTM for sentiment analysis.

## 4.1 RESULTS OVERVIEW

Table 2 (a) lists F1 scores of all relation extraction models. Full results including precision and recall can be found in Appendix A.4. We observe that our proposed NExT consistently outperforms all baseline models in low-resource setting. Also, we found that, (1) directly applying logical forms to unlabeled data results in poor performance. We notice that this method achieves high precision but low recall. Based on our observation of the collected dataset, this is because people tend to use detailed and specific constraints in an NL explanation to ensure they cover all aspects of the instance. As a result, those instances that satisfy the constraints are correctly labeled in most cases, and thus the precision is high. Meanwhile, generalization ability is compromised, and thus the recall is low. (2) Compared to its downstream classifier baseline (BiLSTM+ATT with $\mathcal{S}_a$), NExT achieves 4.2% F1 improvement in absolute value on TACRED, and 5.5% on SemEval. This validates that the expansion of rule coverage by NExT is effective and is providing useful information to classifier training. (3) Performance gap further widens when we take annotation efforts into account. The annotation time for $\mathcal{E}$ and $\mathcal{S}_l$ are equivalent; but the performance of BiLSTM+ATT significantly degrades with fewer instances in $\mathcal{S}_l$. (4) Results of semi-supervised methods are unsatisfactory. This may be explained with difference between underlying data distribution of $\mathcal{S}_a$ and $\mathcal{S}_u$.

Table 2 (b) lists the performances of all sentiment analysis models. The observations are similar to those of relation extraction, which strengthens our conclusions and validates the capability of NExT.

## 4.2 PERFORMANCE ANALYSIS

|  | TACRED | SemEval | Restaurant | Laptop |
|---|---|---|---|---|
| Full NExT | 45.6±0.4 | 63.5±1.0 | 75.8±0.8 | 62.8±1.9 |
| No counting | 44.6±0.9 | 63.2±0.7 | 75.6±0.8 | 62.4±1.9 |
| No matching | 41.8±1.1 | 54.6±1.2 | 71.2±0.4 | 57.0±2.7 |
| No $L_{sim}$ | 42.5±1.0 | 56.2±2.9 | 70.7±0.8 | 59.4±0.7 |
| No $L_{find}$ | 43.2±1.3 | 60.2±0.9 | 70.0±3.5 | 58.1±2.8 |

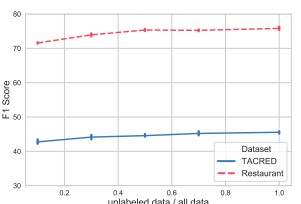

Table 3: Ablation studies on modules of NExT and losses of string matching module. F1 score on the test set is reported. We remove soft counting module (No counting) and string matching module (No matching) by only allowing them to give 0/1 results.

Figure 4: NExT's performance w.r.t. number of unlabeled instances.

**Effectiveness of softening logical rules.** As shown in Table 3, we conduct ablation studies on TA-CRED and Restaurant. We remove two modules that support soft logic (by only allowing them to give 0/1 outputs) to see how much does rule softening help in our framework. Both soft counting module and string matching module contribute to the performance of NExT. It can be easily concluded that string matching module plays a vital role. Removing it leads to significant performance drops, which demonstrates the effectiveness of generalizing when applying logical forms. Besides, we examine the impact brought by $L_{sim}$ and $L_{find}$. Removing them severely hurts the performance, indicating the importance of semantic learning when performing fuzzy matching.

**Performance with different amount of unlabeled data.** To investigate how our NExT's performance is affected by the amount of unlabeled data, we randomly sample 10%, 30%, 50% and 70% of the original unlabeled dataset to do the experiments. As illustrated in Fig. 4, our NExT benefits from larger amount of unlabeled data. We attribute it to high accuracy of logical forms converted from explanations.

**Superiority of explanations in data efficiency.** In the real world, with limited human-power, there is a question of whether it is more valuable to spend time explaining existing annotations than just annotating more labels. To answer this question, we conduct experiments on Performance v.s. Time on TACRED dataset. We compare the results of a supervised classfier with only labels as input and our NExT with both labels and explanations annotated using the same annotation time. The results are listed in Figure 5, from which we can see that NExT achieves higher performance while labeling speed reduces by half.

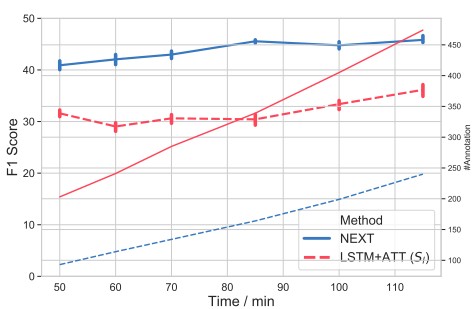

Figure 5: Performance of NExT v.s. traditional supervised method. Blue line denotes NExT and dashed line denotes annotating numbers, normal line means performance. Red line denotes traditional supervised method, and dashed line means performance, normal line means annotating numbers.

**Performance with different number of explanations.** From Fig. 6 , one can clearly observe that all approaches benefit from more labeled data. Our NExT outperforms all other baselines by a large margin, which indicates the effectiveness of leveraging knowledge embedded in NL explanations. We can also see that, the performance of NExT with 170 explanations on TACRED equals to about 2500 labeled data using traditional supervised method. Results of Restaurant also have the same trend, which strengthens our conclusion. Besides Fig. 6, we conduct more experiments for this ablation study and make 4 results tables, see Appendix A.5 for details.

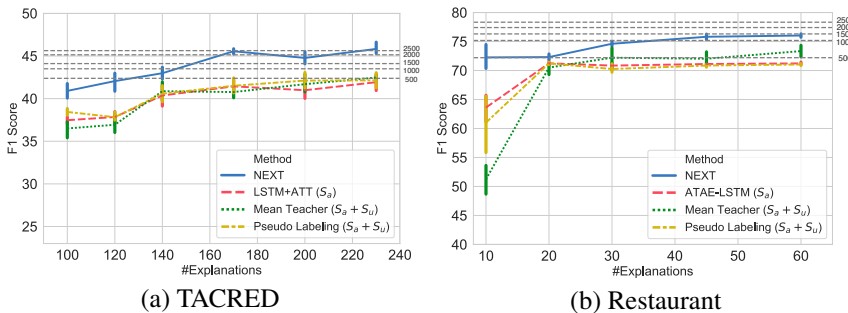

(a) TACRED    (b) Restaurant

Figure 6: **Performance with different number of explanations**. We choose supervised semi-supervised baselines for comparison. We did experiments on TACRED and Restaurant. Gray dashed lines mean the performance with the corresponding labeled data.

### 4.3 ADDITIONAL EXPERIMENT ON MULTI-HOP REASONING

To further test the capability of NExT in downstream tasks, we apply it to WIKIHOP (Welbl et al., 2018) country task by fusing NExT-matched facts into baseline model NLPROLOG (Weber et al., 2019). For a brief introduction, WIKIHOP country task requires a model to select the correct candidate *ENT-Y* for question "Which country is *ENT-X* in?" given a list of support sentences. As part of dataset design, the correct answer can only be found by reasoning over multiple support sentences.

NLPROLOG is a model proposed for WIKIHOP. It first extracts triples from support sentences and treats the masked sentence as the relation between two entities. For example, "Socrate was born in Athens" is converted to (Socrate, "*ENT1* was born in *ENT2*", Athens), where "*ENT1* was born in *ENT2*" is later embedded by SENT2VEC (Pagliardini et al., 2018) to represent the relation between *ENT1* and *ENT2*. Triples extracted from supporting sentences are fed into a Prolog reasoner which will do backward chaining and reasoning to arrive at the target statement country(*ENT-X*, *ENT-Y*). We refer readers to (Weber et al., 2019) for in-depth introduction of NLProlog NLPROLOG.

Fig. 7 shows how the framework in Fig. 2 is adjusted to suit NLPROLOG. We manually choose 3 predicates (i.e., located_in, capital_of, next_to) and annotate 21 support sentences with natural language explanation. We get 103 strictly-matched facts ($\mathcal{S}_a$) and 1407 NExT-matched facts ($\mathcal{S}_u$) among the 128k unlabeled QA support sentences. Additionally, we manually write 5 rules about these 3 predicates for the Prolog solver, e.g. located_in(X,Z) ← located_in(X,Y) ∧ located_in(Y,Z).

Results are listed in Table 4. From the result we observe that simply adding the 103 strictly-matched facts is not making notable improvement. However, with the help of NExT, a larger number of structured facts are recognized from support sentences, so that external knowledge from only 21 explanations and 5 rules improve the accuracy by 1 point. This observation validates NExT's capability in low resource setting and highlight its potential when applied to downstream tasks.

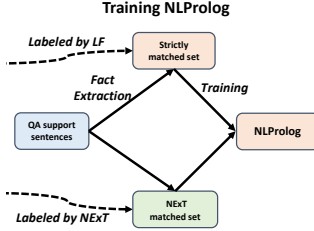

Figure 7: Adjusting NExT Framework (Fig. 2) for NLPRO-LOG.

|  | $|\mathcal{S}_a|$ | $|\mathcal{S}_u|$ | Accuracy |
|---|---|---|---|
| NLProlog (published code) | 0 | 0 | 74.57 |
| + $\mathcal{S}_a$ | 103 | 0 | 74.40 |
| + $\mathcal{S}_u$ (confidence >0.3) | 103 | 340 | 74.74 |
| + $\mathcal{S}_u$ (confidence >0.2) | 103 | 577 | 75.26 |
| + $\mathcal{S}_u$ (confidence >0.1) | 103 | 832 | **75.60** |

Table 4: Performance of NLPROLOG when extracted facts are used as input. Average accuracy over 3 runs is reported. NL-PROLOG empowered by 21 natural language explanations and 5 hand-written rules achieves 1% gain in accuracy.

## 5 RELATED WORK

**Leveraging natural language for training classifiers.** Supervision in the form of natural language has been explored by many works. Srivastava et al. (2017) first demonstrate the effectiveness of NL explanations. They proposed a joint concept learning and semantic parsing method for classification problems. However, the method is very limited in that it is not able to use unlabeled data. To address this issue, Hancock et al. (2018) propose to parse the NL explanations into labeling functions and then use data programming to handle the conflict and enhancement between different labeling functions. Camburu et al. (2018) extend Stanford Natural Language Inference dataset with NL explanations and demonstrate its usefulness for various goals for training classifiers. Andreas et al. (2016) explore decomposing NL questions into linguistic substructures for learning collections of neural modules which can be assembled into deep networks. Hu et al. (2019) explore using NL instructions as compositional representation of actions for hierarchical decision making. The substructure of an instruction is summarized as a latent plan, which is then executed by another model. Rajani et al. (2019) train a language model to automatically generate NL explanations that can be used during training and inference for the task of commonsense reasoning.

**Weakly-supervised learning.** Our work is relevant to weakly-supervised learning. Traditional systems use handcrafted rules (Hearst, 1992) or automatically learned rules (Agichtein & Gravano, 2000; Batista et al., 2015) to take a rule-based approach. Hu et al. (2019) incorporate human knowledge into neural networks by using a teacher network to teach the classifier knowledge from rules and train the classifier with labeled data. Li et al. (2018) parse regular expression to get action trees as a classifier that are composed of neural modules, so that essentially training stage is just a process of learning human knowledge. Meanwhile, if we regard those data that are exactly matched by rules as labeled data and the remaining as unlabeled data, we can apply many semi-supervised models such as self learning (Rosenberg et al., 2005), mean-teacher (Tarvainen & Valpola, 2017), and semi-supervised VAE (Xu et al., 2017). However, These models turn out to be ineffective in rule-labeled data or explanation-labeled data due to potentially large difference in label distribution. The data sparsity is also partially solved by distant supervision (Mintz et al., 2009; Surdeanu et al., 2012). They rely on knowledge bases (KBs) to annotate data. However, the methods introduce a lot of noises, which severely hinders the performance. Liu et al. (2017) instead propose to conduct relation extraction using annotations from heterogeneous information source. Again, predicting true labels from noisy sources is challenging.

## 6 CONCLUSION

In this paper, we presented NExT, a framework that augments sequence classification by exploiting NL explanations as supervision under a low resource setting. We tackled the challenges of modeling the compositionality of NL explanations and dealing with the linguistic variants. Four types of modules were introduced to generalize the different types of actions in logical forms, which substantially increase the coverage of NL explanations. A joint training algorithm was proposed to utilize information from both labeled dataset and unlabeled dataset. We conducted extensive experiments on several datasets and proved the effectiveness of our model. Future work includes extending NExT to sequence labeling tasks and building a cross-domain semantic parser for NL explanations.

### ACKNOWLEDGMENTS

This research is based upon work supported in part by the Office of the Director of National Intelligence (ODNI), Intelligence Advanced Research Projects Activity (IARPA), via Contract No. 2019-19051600007, NSF SMA 18-29268, and Snap research gift. The views and conclusions contained herein are those of the authors and should not be interpreted as necessarily representing the official policies, either expressed or implied, of ODNI, IARPA, or the U.S. Government. We would like to thank all the collaborators in USC INK research lab for their constructive feedback on the work.

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

## A  APPENDIX

### A.1  PREDICATES

Following Srivastava et al. (2017), we first compile a domain lexicon that maps each word to its syntax and logical predicate. Table 5 lists some frequently used predicates in our parser, descriptions about their function and modules they belong to.

| Predicate | Description | Module |
|---|---|---|
| Because, Separator
ArgX, ArgY, Arg
Int, Token, String
True, False | Basic conjunction words
Subject, object or aspect term in each task
Primitive data types
Boolean operators | *None* |
| And, Or, Not, Is, Occur | Logical operators that aggregate matching scores | *Logical Calculation Module* |
| Left, Right, Between, Within

NumberOf | Return True if one string is left/right/between/within
some range of the other string
Return the number of words in a given range | *Deterministic Function* |
| AtMost, AtLeast, Direct,
MoreThan, LessThan, Equals | Counting (distance) constraints | *Soft Counting Module* |
| Word, Contains, Link | Return a matching score sequence for a sentence and a query | *String Matching Module* |

Table 5: Frequently used predicates

### A.2  EXAMPLES FOR COLLECTED EXPLANATIONS.

**TACRED**

*OBJ-ORGANIZATION coach SUBJ-PERSON insisted he would put the club 's 2-0 defeat to Palermo firmly behind him and move forward*

```
(Label) per:employee_of
```

```
(Explanation) there is only one word "coach" between SUBJ and OBJ
```

*Officials in Mumbai said that the two suspects , David Coleman Headley , an American with links of Pakistan , and SUBJ-PERSON , who was born in Pakistan but is a OBJ-NATIONALITY citizen , both visited Mumbai and several other Indian cities in before the attacks , and may have visited some of the sites that were attacked*

```
(Label) per:origin
```

```
(Explanation) the words "is a" appear right before OBJ-NATIONALITY
and the word "citizen" is right after OBJ-NATIONALITY
```

**SemEval 2010 Task 8**

*The SUBJ-O is caused by the OBJ-O of UV radiation by the oxygen and ozone*

```
(Label) Cause-Effect(e2,e1)
```

```
(Explanation) the phrase "is caused by the" occurs between SUBJ
and OBJ and OBJ follows SUBJ
```

*SUBJ-O are parts of the OBJ-O OBJ-O disregarded by the compiler*

```
(Label) Component-Whole(e1,e2)
```

```
(Explanation) the phrase "are parts of the" occurs between SUBJ
and OBJ and OBJ follows SUBJ
```

**SemEval 2014 Task 4 - restaurant**

*I am relatively new to the area and tried Pick a bgel on 2nd and was disappointed with the service and I thought the food was overated and on the pricey side (Term: food)*

```
(Label) negative
```

```
(Explanation) the words "overated" is within 2 words after term
```

*The decor is vibrant and eye-pleasing with several semi-private boths on the right side of the dining hall, which are great for a date (Term: decor)*

```
(Label) positive
```

```
(Explanation) the term is followed by "vibrant" and "eye-pleasing"
```

**SemEval 2014 Task 4 - laptop**

*It's priced very reasonable and works very well right out of the box. (Term: works)*

```
(Label) positive
```

```
(Explanation) the words "very well" occur directly after the term
```

*The DVD drive randomly pops open when it is in my backpack as well, which is annoying (Term: DVD drive)*

```
(Label) negative
```

```
(Explanation) the word "annoying" occurs after term
```

## A.3 IMPLEMENTATION DETAILS

We use 300-dimensional word embeddings pre-trained by GloVe (Pennington et al., 2014). The dropout rate is 0.96 for word embeddings and 0.5 for sentence encoder. The hidden state size of the encoder and attention layer is 300 and 200 respectively. We choose Adagrad as the optimizer and the learning rate for joint model learning is 0.5.

For TACRED, we set the learning rate to 0.1 in the pretraining stage. The total epochs for pretraining are 10. The weight for $L_{sim}$ is set to 0.5. The batch size for pretraining is set to 100. For training the classifier, the batch size for labeled data and unlabeled data is 50 and 100 respectively, the weight $\alpha$ for $L_u$ is set to 0.7, the weight $\beta$ for $L_{string}$ is set to 0.2, the weight $\gamma$ for $L_{sim}$ is set to 2.5.

For SemEval 2010 Task 8, we set the learning rate to 0.1 in the pretraining stage. The total epochs for pretraining are 10. The weight for $L_{sim}$ is set to 0.5. The batch size for pretraining is set to 10. For training the classifier, the batch size for labeled data and unlabeled data is 50 and 100 respectively, the weight $\alpha$ for $L_u$ is set to 0.5, the weight $\beta$ for $L_{string}$ is set to 0.1, the weight $\gamma$ for $L_{sim}$ is set to 2.

For two datasets in SemEval 2014 Task 4, we set the learning rate to 0.5 in the pretraining stage. The total epochs for pretraining are 20. The weight for $L_{sim}$ is set to 5. The batch size for pretraining is set to 20. For training the classifier, the batch size for labeled data and unlabeled data is 10 and 50 respectively, the weight $\alpha$ for $L_u$ is set to 0.5, the weight $\beta$ for $L_{string}$ is set to 0.1, the weight $\gamma$ for $L_{sim}$ is set to 2. For ATAE-LSTM, we set hidden state of attention layer to be 300 dimension.

## A.4 FULL RESULTS FOR MAIN EXPERIMENTS

The full results for relation extraction and sentiment analysis are listed in Table 6 and Table 7 respectively.

| | TACRED | | | SemEval | | |
|---|---|---|---|---|---|---|
| Metric | Precision | Recall | F1 | Precision | Recall | F1 |
| LF ($\mathcal{E}$) | **83.21** | 13.56 | 23.33 | **83.19** | 21.26 | 33.86 |
| CBOW-GloVe ($\mathcal{R} + \mathcal{S}$) | 28.2±0.7 | **44.9±0.9** | 34.6±0.4 | 46.8±1.3 | 51.2±2.2 | 48.8±1.1 |
| PCNN ($\mathcal{S}_a$) | 43.8±1.6 | 28.9±1.1 | 34.8±0.9 | 51.5±1.9 | 35.2±1.4 | 41.8±1.2 |
| PA-LSTM ($\mathcal{S}_a$) | 44.4±2.9 | 38.7±2.2 | 41.3±0.8 | 59.9±2.4 | 54.9±2.2 | 57.3±1.5 |
| BiLSTM+ATT ($\mathcal{S}_a$) | 43.8±2.0 | 39.4±2.6 | 41.4±1.0 | 60.0±2.1 | 56.2±1.3 | 58.0±1.6 |
| BiLSTM+ATT ($\mathcal{S}_l$) | 42.8±2.6 | 23.8±2.4 | 30.4±1.4 | 54.7±1.0 | 53.6±1.2 | 54.1±1.0 |
| Data Programming ($\mathcal{E} + \mathcal{S}$) | 45.9±2.8 | 23.3±2.6 | 30.8±2.4 | 51.3±3.5 | 38.8±4.2 | 43.9±2.4 |
| Self Training ($\mathcal{S}_a + \mathcal{S}_u$) | 45.9±2.3 | 38.4±2.7 | 41.7±1.5 | 57.3±2.1 | 53.3±0.9 | 55.2±0.8 |
| Pseudo Labeling ($\mathcal{S}_a + \mathcal{S}_u$) | 44.5±1.5 | 38.9±1.6 | 41.5±1.2 | 53.7±2.6 | 53.4±2.2 | 53.5±1.2 |
| Mean Teacher ($\mathcal{S}_a + \mathcal{S}_u$) | 39.2±1.7 | 42.6±1.8 | 40.8±0.9 | 60.8±1.9 | 51.9±1.2 | 56.0±1.1 |
| Mean Teacher ($\mathcal{S}_l + \mathcal{S}_{lu}$) | 28.3±5.7 | 25.4±5.8 | 25.9±2.2 | 53.1±3.8 | 51.6±2.4 | 52.2±0.7 |
| DualRE ($\mathcal{S}_a + \mathcal{S}_u$) | 38.8±4.7 | 28.6±2.9 | 32.6±0.7 | 64.5±0.7 | 59.2±2.0 | 61.7±0.9 |
| NExT ($\mathcal{E} + \mathcal{S}$) | 49.2±0.9 | 42.4±1.3 | **45.6±0.4** | 66.3±1.4 | **61.0±2.2** | **63.5±1.0** |

Table 6: Full results as supplement to Table 2(a)

| | Restaurant | | | Laptop | | |
|---|---|---|---|---|---|---|
| Metric | Precision | Recall | F1 | Precision | Recall | F1 |
| LF ($\mathcal{E}$) | **86.5** | 4.0 | 7.7 | **90.0** | 7.1 | 13.1 |
| CBOW-GloVe ($\mathcal{R} + \mathcal{S}$) | 62.8±2.8 | 75.3±3.1 | 68.5±2.9 | 53.4±1.1 | 72.6±1.5 | 61.5±1.3 |
| PCNN ($\mathcal{S}_a$) | 67.1±2.1 | 79.0±1.8 | 72.6±1.2 | 53.1±1.0 | 71.4±1.1 | 60.9±1.1 |
| ATAE-LSTM ($\mathcal{S}_a$) | 65.1±0.4 | 78.4±0.6 | 71.1±0.4 | 49.0±3.1 | 66.0±4.4 | 56.2±3.6 |
| ATAE-LSTM ($\mathcal{S}_l$) | 65.3±0.5 | 78.9±0.5 | 71.4±0.5 | 48.9±1.5 | 55.6±2.4 | 52.0±1.4 |
| Data Programming ($\mathcal{E} + \mathcal{S}$) | 65.0±0.0 | 78.8±0.0 | 71.2±0.0 | 53.4±0.1 | 72.5±0.1 | 61.5±0.1 |
| Self Training ($\mathcal{S}_a + \mathcal{S}_u$) | 65.3±0.7 | 78.4±0.9 | 71.2±0.5 | 50.1±1.8 | 67.7±2.4 | 57.6±2.1 |
| Pseudo Labeling ($\mathcal{S}_a + \mathcal{S}_u$) | 64.9±0.5 | 78.0±0.6 | 70.9±0.4 | 50.4±1.6 | 68.4±2.3 | 58.0±1.9 |
| Mean Teacher ($\mathcal{S}_a + \mathcal{S}_u$) | 68.8±2.2 | 75.7±3.9 | 72.0±1.5 | 54.4±1.7 | 72.3±4.0 | 62.1±2.3 |
| Mean Teacher ($\mathcal{S}_l + \mathcal{S}_{lu}$) | 68.3±0.8 | 81.0±0.4 | 74.1±0.4 | 55.0±4.1 | 70.3±3.3 | 61.7±3.7 |
| NExT ($\mathcal{E} + \mathcal{S}$) | 69.6±0.9 | **83.3±1.8** | **75.8±0.8** | 54.6±1.6 | **73.9±2.3** | **62.8±1.9** |

Table 7: Full results as supplement to Table 2(b)

## A.5 Performance with Different Number of Explanations

As a supplement to Fig. 6, we show the full experimental results with different number of explanations as input in Table 8,9,10,11. Results show that our model achieves best performance compared with baseline methods.

## A.6 Model-agnostic

Our framework is model-agnostic as it can be integrated with any downstream classifier. We conduct experiments on SA Restaurant dataset with 45 and 75 explanations using BERT as downstream classifier, and the results are summarized in Table 12. Results show that our model still outperforms baseline methods when BERT is incorporated. We observe the performance of NExT is approaching the upper bound 85% (by feeding all data to BERT), with only 75 explanations, which again demonstrates the annotation efficiency of NExT.

## A.7 Case Study on String Matching Module.

String matching module plays a vital role in NExT. The matching quality greatly influences the accuracy of pseudo labeling. In Fig. 8, we can see that keyword *chief executive of* is perfectly aligned with *executive director of* in the sentence, which demonstrates the effectiveness of string matching module in capturing semantic similarity.

| | TACRED 130 | | | TACRED 100 | | |
|---|---|---|---|---|---|---|
| Metric | Precision | Recall | F1 | Precision | Recall | F1 |
| LF ($\mathcal{E}$) | **83.5** | 12.8 | 22.2 | **85.2** | 11.8 | 20.7 |
| CBOW-GloVe ($\mathcal{R} + \mathcal{S}$) | 26.0±2.3 | **39.9±5.0** | 31.2±0.5 | 24.4±1.3 | **41.7±3.7** | 30.7±0.1 |
| PCNN ($\mathcal{S}_a$) | 41.8±2.7 | 28.8±1.8 | 34.1±1.1 | 28.2±3.4 | 22.2±1.3 | 24.8±1.9 |
| PA-LSTM ($\mathcal{S}_a$) | 44.9±1.7 | 33.5±2.9 | 38.3±1.3 | 39.9±2.1 | 38.2±1.1 | 39.0±1.3 |
| BiLSTM+ATT ($\mathcal{S}_a$) | 40.1±2.6 | 36.2±3.4 | 37.9±1.1 | 36.1±0.4 | 37.6±3.0 | 36.8±1.4 |
| BiLSTM+ATT ($\mathcal{S}_l$) | 35.0±9.0 | 25.4±1.6 | 28.9±2.7 | 43.3±2.2 | 23.1±3.3 | 30.0±3.1 |
| Self Training ($\mathcal{S}_a + \mathcal{S}_u$) | 43.6±3.3 | 35.1±2.1 | 38.7±0.0 | 41.9±5.9 | 32.0±7.4 | 35.5±2.5 |
| Pseudo Labeling ($\mathcal{S}_a + \mathcal{S}_u$) | 44.2±1.9 | 34.2±1.9 | 38.5±0.6 | 39.7±2.0 | 34.9±3.3 | 37.1±1.5 |
| Mean Teacher ($\mathcal{S}_a + \mathcal{S}_u$) | 38.8±0.9 | 35.6±1.3 | 37.1±0.5 | 37.4±4.0 | 37.4±0.2 | 37.3±2.0 |
| Mean Teacher ($\mathcal{S}_l + \mathcal{S}_{lu}$) | 21.1±3.3 | 28.7±1.8 | 24.2±1.8 | 17.5±4.7 | 18.4±.59 | 17.9±5.0 |
| DualRE ($\mathcal{S}_a + \mathcal{S}_u$) | 34.9±3.6 | 30.5±2.3 | 32.3±1.0 | 40.6±4.3 | 19.1±1.5 | 25.9±0.6 |
| Data Programming ($\mathcal{E} + \mathcal{S}$) | 34.3±16.1 | 18.7±1.4 | 23.5±4.9 | 43.5±2.3 | 15.0±2.3 | 22.2±2.4 |
| NEXT ($\mathcal{E} + \mathcal{S}$) | 45.3±2.4 | 39.2±0.3 | **42.0±1.1** | 43.9±3.7 | 36.2±1.9 | **39.6±0.5** |

Table 8: TACRED results on 130 explanations and 100 explanations

| | SemEval 150 | | | SemEval 100 | | |
|---|---|---|---|---|---|---|
| Metric | Precision | Recall | F1 | Precision | Recall | F1 |
| LF ($\mathcal{E}$) | **85.1** | 17.2 | 28.6 | **90.7** | 9.0 | 16.4 |
| CBOW-GloVe ($\mathcal{R} + \mathcal{S}$) | 44.8±1.9 | 48.6±1.5 | 46.6±1.1 | 36.0±1.4 | 40.2±2.0 | 37.9±0.1 |
| PCNN ($\mathcal{S}_a$) | 49.1±3.9 | 36.1±2.4 | 41.5±1.4 | 43.3±1.4 | 27.9±1.0 | 33.9±0.3 |
| PA-LSTM ($\mathcal{S}_a$) | 58.0±1.2 | 52.5±0.4 | 55.1±0.5 | 55.2±1.7 | 37.7±0.8 | 44.8±0.8 |
| BiLSTM+ATT ($\mathcal{S}_a$) | 59.2±0.4 | 53.7±1.8 | 56.3±0.8 | 54.9±5.0 | 40.5±0.9 | 46.5±1.3 |
| BiLSTM+ATT ($\mathcal{S}_l$) | 47.6±2.6 | 42.0±2.3 | 44.6±2.5 | 43.7±2.6 | 37.6±5.0 | 40.3±3.7 |
| Self Training ($\mathcal{S}_a + \mathcal{S}_u$) | 53.4±4.3 | 47.5±2.9 | 50.1±1.1 | 53.2±2.3 | 34.2±2.2 | 41.6±1.4 |
| Pseudo Labeling ($\mathcal{S}_a + \mathcal{S}_u$) | 55.3±4.5 | 51.0±2.3 | 53.0±1.5 | 47.4±4.6 | 39.9±3.9 | 43.1±0.6 |
| Mean Teacher ($\mathcal{S}_a + \mathcal{S}_u$) | 61.8±4.0 | 49.1±2.6 | 54.6±0.2 | 58.5±1.9 | 41.8±2.6 | 48.7±1.4 |
| Mean Teacher ($\mathcal{S}_l + \mathcal{S}_{lu}$) | 40.6±2.0 | 31.2±4.5 | 35.2±3.6 | 32.7±3.0 | 25.6±3.1 | 28.6±2.2 |
| DualRE ($\mathcal{S}_a + \mathcal{S}_u$) | 61.7±3.0 | 56.1±3.0 | 58.8±3.0 | 61.6±1.7 | 39.7±1.9 | 48.3±1.5 |
| Data Programming ($\mathcal{E} + \mathcal{S}$) | 50.9±10.8 | 27.0±0.8 | 35.0±3.2 | 28.0±4.1 | 17.4±5.5 | 21.0±3.4 |
| NEXT ($\mathcal{E} + \mathcal{S}$) | 68.5±1.6 | **60.0±1.7** | **63.7±0.8** | 60.2±1.8 | **53.5±0.7** | **56.7±1.1** |

Table 9: SemEval results on 150 explanations and 100 explanations

| | Laptop 55 | | | Laptop 70 | | |
|---|---|---|---|---|---|---|
| Metric | Precision | Recall | F1 | Precision | Recall | F1 |
| LF ($\mathcal{E}$) | **90.8** | 9.2 | 16.8 | **89.4** | 9.2 | 16.8 |
| CBOW-GloVe ($\mathcal{R} + \mathcal{S}$) | 53.7±0.2 | 72.9±0.2 | 61.8±0.2 | 53.6±0.3 | 72.4±0.2 | 61.6±0.2 |
| PCNN ($\mathcal{S}_a$) | 53.5±3.3 | 71.0±3.6 | 61.0±3.2 | 55.6±1.9 | 74.1±1.9 | 63.5±1.5 |
| ATAE-LSTM ($\mathcal{S}_a$) | 53.5±0.4 | 71.9±2.2 | 61.3±1.0 | 53.7±1.2 | 72.9±1.8 | 61.9±1.5 |
| ATAE-LSTM ($\mathcal{S}_l$) | 48.3±1.0 | 59.5±5.0 | 53.2±2.2 | 54.1±1.4 | 61.1±3.0 | 57.4±2.1 |
| Self Training ($\mathcal{S}_a + \mathcal{S}_u$) | 51.3±2.6 | 68.6±2.7 | 58.7±2.6 | 51.2±1.4 | 68.6±2.2 | 58.7±1.6 |
| Pseudo Labeling ($\mathcal{S}_a + \mathcal{S}_u$) | 51.8±1.7 | 70.3±2.3 | 59.7±1.9 | 52.4±0.8 | 70.9±1.5 | 60.3±1.0 |
| Mean Teacher ($\mathcal{S}_a + \mathcal{S}_u$) | 55.1±0.9 | 74.1±1.6 | 63.2±1.1 | 55.9±3.3 | 73.0±2.6 | 63.2±1.7 |
| Mean Teacher ($\mathcal{S}_l + \mathcal{S}_{lu}$) | 55.5±2.5 | 69.3±2.8 | 61.6±2.2 | 58.0±0.7 | 73.2±1.5 | 64.7±1.0 |
| Data Programming ($\mathcal{E} + \mathcal{S}$) | 53.4±0.0 | 72.6±0.0 | 61.5±0.0 | 53.5±0.1 | 72.5±0.1 | 61.6±0.1 |
| NEXT ($\mathcal{E} + \mathcal{S}$) | 56.3±1.3 | **75.9±2.5** | **64.6±1.7** | 56.9±0.2 | **77.1±0.6** | **65.5±0.3** |

Table 10: Laptop results on 55 explanations and 70 explanations

| | Restaurant 60 | | | Restaurant 75 | | |
|---|---|---|---|---|---|---|
| Metric | Precision | Recall | F1 | Precision | Recall | F1 |
| LF ($\mathcal{E}$) | **86.0** | 3.8 | 7.4 | **85.4** | 6.8 | 12.6 |
| CBOW-GloVe ($\mathcal{R} + \mathcal{S}$) | 63.7±2.3 | 75.6±1.3 | 69.1±1.9 | 64.1±1.3 | 76.6±0.1 | 69.8±0.7 |
| PCNN ($\mathcal{S}_a$) | 67.0±0.9 | 81.0±1.0 | 73.3±0.9 | 68.4±0.1 | **82.8±0.3** | 74.9±0.2 |
| ATAE-LSTM ($\mathcal{S}_a$) | 65.2±0.6 | 78.5±0.2 | 71.2±0.3 | 64.7±0.4 | 78.3±0.4 | 70.8±0.4 |
| ATAE-LSTM ($\mathcal{S}_l$) | 67.0±1.5 | 79.5±1.2 | 72.7±1.0 | 66.6±2.0 | 78.5±1.4 | 72.1±0.6 |
| Self Training ($\mathcal{S}_a + \mathcal{S}_u$) | 65.2±0.2 | 78.7±0.5 | 71.3±0.2 | 65.7±1.1 | 77.2±1.1 | 71.0±0.1 |
| Pseudo Labeling ($\mathcal{S}_a + \mathcal{S}_u$) | 64.9±0.6 | 77.8±1.0 | 70.8±0.3 | 64.9±0.9 | 77.8±1.2 | 70.7±1.0 |
| Mean Teacher ($\mathcal{S}_a + \mathcal{S}_u$) | 68.8±2.3 | 76.0±2.2 | 72.2±1.3 | 73.3±3.5 | 79.2±3.8 | 76.0±1.2 |
| Mean Teacher ($\mathcal{S}_l + \mathcal{S}_{lu}$) | 69.0±0.8 | 82.0±1.1 | 74.9±0.7 | 69.2±0.7 | 82.6±0.6 | 75.3±0.6 |
| Data Programming ($\mathcal{E} + \mathcal{S}$) | 65.0±0.0 | 78.8±0.1 | 71.2±0.0 | 65.0±0.0 | 78.8±0.0 | 71.2±0.0 |
| NEXT ($\mathcal{E} + \mathcal{S}$) | 71.0±1.4 | **82.8±1.1** | **76.4±0.4** | 71.9±1.5 | **82.8±1.9** | **76.9±0.7** |

Table 11: Restaurant results on 60 explanations and 75 explanations

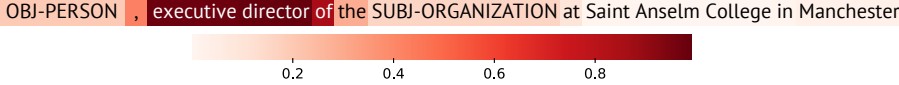

Figure 8: Heatmap for keyword *chief executive of* and sentence *OBJ-PERSON, executive director of the SUBJ-ORGANIZATION at Saint Anselm College in Manchester*. Results show that our string matching module can successfully grasp relevant words.

| | 45 | 75 |
|---|---|---|
| ATAE-LSTM ($\mathcal{S}_a$) | 79.9 | 80.6 |
| Self Training ($\mathcal{S}_a + \mathcal{S}_u$) | 80.9 | 81.1 |
| Pseudo Labeling ($\mathcal{S}_a + \mathcal{S}_u$) | 78.7 | 81.0 |
| Mean Teacher ($\mathcal{S}_a + \mathcal{S}_u$) | 79.3 | 79.8 |
| NExT ($\mathcal{E} + \mathcal{S}$) | **81.4** | **82.0** |

Table 12: BERT experiments on Restaurant dataset using 45 and 75 explanations

