# OpenReview forum: "Learning from Explanations with Neural Execution Tree"
_ICLR.cc/2020/Conference — Accept (Poster)_

### Official Review · AnonReviewer1 · 2019-10-22
**Official Blind Review #1**

**Rating:** 8

**Review:**


This paper explores using natural language explanations as auxiliary training data for NLP tasks. It first transforms natural language expressions into a logical form through CCG, and then use a neural module network architecture to label data instances. Experimental analyses are conducted on two tasks -- relation extraction and sentiment analysis, showing that the proposed approach outperforms previous work that incorporates explanations as training data.

Overall, the paper addresses an important issue of how to utilize human explanations as additional supervision source for NLP tasks and shows promising results. Hence, I believe the paper is above the acceptance threshold, and recommend for weak acceptance.

I had concerns on the cost of collecting human explanations, since they are non-trivial to collect. However, the authors provided convincing arguments regarding the data annotation cost in their response, so I do not think this is a major limitation of the method.

However, I would also like to note that the paper has a few limitations. The proposed is based on semantic parsing of the natural language explanations into logical forms and is therefore inherently limited by the representation power of symbolic and logical representations. In the two examples shown in Figure 1, the human explanations are very simple and have limited variety, so it is relatively easy to represent them in logical forms. However, on many NLP tasks (such as question answering), the human explanations (in natural language) may often be complicated and difficult to be represented in CCG. Therefore, it is unclear whether the proposed approach can be easily generalized to other tasks.


**Experience Assessment:**

I have read many papers in this area.

**Review Assessment: Checking Correctness Of Derivations And Theory:**

N/A

**Review Assessment: Checking Correctness Of Experiments:**

I carefully checked the experiments.

**Review Assessment: Thoroughness In Paper Reading:**

I read the paper at least twice and used my best judgement in assessing the paper.

---

> ### Author Response · Authors · 2019-11-13
> **Response to reviewer 1**
>
> We really appreciate your comments and valuable suggestions for improving this work!
>
> -------------------------------
> Q1: It is unclear whether the proposed methods can be generalized to other tasks (such as question answering):
>
> A1: Indeed, it is challenging to generalize our framework to more complex tasks such as Question Answering. Still, we believe this is a promising direction to explore.
>
> In this paper, we've made some attempts by applying NMET to a multi-hop multiple-choice Question Answering task (WikiHop in section 4.3) and it's making promising improvements. From the results, simply adding 100+ strictly-matched facts is not making a notable improvement, however, with the help of NMET, external knowledge from only 21 NL explanations and 5 rules help recognize a large number of structured facts and improves the performance.
>
> Extending our framework to span-based QA (e.g. SQuAD) is a research problem that we’re recently tackling, and we would like to show the following example.
>
> Question: How is packet switching characterized?
> Context: Circuit switching is characterized by a fee per unit of connection time, even when no data is transferred, while packet switching may be characterized by a fee per unit of information transmitted,
> Answer: by a fee per unit of information transmitted.
>
> A good explanation for such a dataset should cover three things: (1) What are the keywords in the question or context? (2) How is this type of question usually answered? (3) How do we locate the answer? For the above example, human annotators can write an NL explanation as follows: In the question, the words "packet switching" and "characterized" are important phrases. The question starts with "How is", so the answer probably starts with "by". The answer is right after "characterized" and the phrase "may be" is between "packet switching" and "characterized".
>
> We can easily adapt our CCG parser and conduct semantic parsing on the above explanation. It can be parsed into the following logical form (a little different from what we defined in this paper):
>
> - Q=Question, S=Sentence
> - Variables: X(NP), Y(VBZ), ANS(PP)
> - Rules1
> @Is(Q, @StartWith(“How is”))
> @Is(X, @Direct(@Right(“How is”)))
> @Is(Y, @Direct(@Right(X)))
> @Is(Q, @EndWith(Y))
> - Rules2
> @Is(X, @AppearIn(S))
> @Is(Y, @AppearIn(S))
> @Is(“may be”, @Between(X,Y))
> - Rules3
> @Is(ANS, @StartWith(“By”))
> - Rules4
> @Is(ANS, @Direct(@Right(Y))
>
> This logical form can then be applied to answer the following question:
>
> Question: How is cheese made?
> Context: Cheese is made the same way — by curdling milk — except the milk is curdled on purpose. Most cheese is made in factories. After milk is poured into big vats, a “starter culture” of bacteria is added to convert the lactose into lactic acid.
>
> To answer this question with the above rule, we first pre-process the question/context into chunks to fill the variables (X, Y) with candidate chunks in the question (e.g. X=cheese, Y=made). We score all candidates with Rule1. If the question matches with Rule1 with high confidence, we further retrieve relevant context with Rule2 in context. We later fill the variable ANS with selected sentence with Rule3. In the end, we evaluate the candidate ANS with Rule4. The evaluation of candidates with Rule1-4 can be done with the proposed NMET framework. The answer to the above instance, "by curdling milk" can be correctly selected.
>
> Alternatively, we can leverage this LF to match context first (with Rule2) and then generate question (with Rule1) and answer (with Rule3-4).
>
> -------------------------------
> Q2: Besides, it is also non-trivial to collect human explanations.
>
> A2: Given that an annotator has viewed the whole instance and gives a label, it should be easy for them to write a corresponding NL explanation explaining how the decision is made. It has been found in previous work that, on average it took the same amount of time to collect 30 explanations as 60 labels ([1]) in relation extraction task. Previous work also found that collecting annotator rationales in the form of highlighted substrings from the sentence only doubles annotation time ([2]). From our experiments, we find that for skilled annotators, the average ratio of (A) the time for only giving a label and (B) the time for giving both a label and an explanation in RE task is 1:2.01 for TACRED and 1:1.99 for SemEval, while in SA task it is 1:2.22 for Laptop and 1:2.30 for Restaurant.
>
> As is discussed in section 4.2, compared with giving labels only, NL explanations with our NMET achieves higher data annotation efficiency (considering both model performance and annotation time).
>
> [1] Training Classifiers with Natural Language Explanations   ACL 2018
> [2] Modeling Annotators: A Generative Approach to Learning from Annotator Rationales   EMNLP 2008

---

### Official Review · AnonReviewer3 · 2019-10-23
**Official Blind Review #3**

**Rating:** 8

**Review:**

Labeling sentences for NLP requires a lot of human effort. In order to tackle this problem, the system proposed in this paper, called NMET, aims at labeling sentences by exploiting the explanations given by humans. First it converts the explanations into logical formulas. This logical formulas are then exploited for partitioning the dataset into two datasets: labeled dataset and unlabeled dataset. Then, NMET relaxes the logical formulas for labeling unlabeled examples by exploiting a neural architecture that uses four modules to deal with different types of predicates.

The paper is pretty clear and well-written. It can be understood even by non-experts. The proposed approach seems technically sound and pretty novel. Moreover, the experimental results show that the proposed system achieves better performances than traditional label-only supervised models.
The only concern that I see in the paper is that the Deterministic Function Module is not explained very well. It is not clear to me its purpose.

[Minor]
Page 3
“the the logical form”


**Experience Assessment:**

I do not know much about this area.

**Review Assessment: Checking Correctness Of Derivations And Theory:**

I did not assess the derivations or theory.

**Review Assessment: Checking Correctness Of Experiments:**

I assessed the sensibility of the experiments.

**Review Assessment: Thoroughness In Paper Reading:**

I made a quick assessment of this paper.

---

> ### Author Response · Authors · 2019-11-13
> **Response to reviewer 3**
>
> We really appreciate your detailed comments and valuable feedback!
>
> > The Deterministic Function Module is not explained very well.
>
> We are sorry for the confusion and unclear explanation about the Deterministic Function Module. Some predicates defined by human annotators, like ‘The phrase *** occurs’, can be softened by our NMET framework to improve the generalization ability. However, there are other predicates that are deterministic, which means they can only be exactly matched, like ‘Left’ and ‘Right’. A string is either right or left of an anchor word. Therefore, the probability it outputs should be either 0 or 1. The Deterministic Function Module deals with all these predicates and outputs a mask sequence, which is fed into the tree structure and combined with other information. We’ll discuss it in detail in our paper.

---

### Official Review · AnonReviewer2 · 2019-10-25
**Official Blind Review #2**

**Rating:** 3

**Review:**

One recent work that comes to mind from ACL 2019: Leveraging Language Models for Commonsense Reasoning (Rajani et al 2019). In that work, they also have human annotators provide explanations (extending the CommonsenseQA dataset), and they show that by training with these explanations, inference is improved even without them. They also train a language model to generate the explanations, and they show that the language model generated explanations improve performance further at inference time. Seems like a reasonable reference to contrast the more structured approach to using explanations like Srivasta et al (2017), Hancock et al (2018), and this work.

I find the method summary beginning with "Human explanations are first converted to machine-actionable logical forms by a semantic parser" until the end of that paragraph to be unnecessary. Actually, as I read it, I find myself asking a lot of questions that get answered below. So I would prefer scrapping that method summary and just getting straight into the Explanation Parsing.

"indicates the the logical form matches" redundant 'the'

I can't find a definition for LF(E) anywhere, and yet LF(E) is present in many tables. I see that E is mentioned to be the explanations, but this is only in the caption of Table 2 even though the symbol is first used in the first paragraph of Section 4.1. I'm assuming LF logical forms applied directly to explanations, but this should be stated explicitly. Can you elaborate on why it is so dominant on precision In Table 6 and 7 of the Appendix, but low on recall, rather than just saying this is expected in Section 4.1?

"For keyword query q, we directly encode it into vector z_q by bi-LSTM and attention." Can you elaborate on how z_q is constructed? Is it the final state of a forward LSTM concatenated with the final state of a backward LSTM?

Why is it so essential that you study the setting in which explanations are low-resource? I'm curious to see what would happen with more explanations.

I am surprised that none of the modules or compared methods include any architecture that use a Transformer or a form of contextualized word vectors (McCann et al 2017, Peters et al 2017, Devlin et al 2018). Is there an explanation for this?

I would prefer to see a larger suite of tested tasks given that each of these datasets is quite small. Would any other tasks from benchmarks like GLUE or SuperGLUE be amenable to your approach? The tasks you've chosen limit the scope of this work and leaves the question of whether it would generally improve across a greater variety of tasks, especially tasks that have seen significant improvement using new methods. Your claim would be much stronger if the explanations were shown to be helpful even to pretrained models like BERT when fine-tuned for a specific task. In particular, it would be interesting to see how the benefits of explanations vary for different kinds of tasks and for different training set sizes.

"In the real world, a more realistic problem is that, with limited human-power, should we just annotate more labels or spend time explaining existing annotations."" I think that you mean "should we" makes it sound like this is a question, but there is no question mark, and it makes more sense as a statement. I propose "...human-power, there is a question of whether it is more valuable to ..."

"Explanations prove to be an very efficient form for data annotation." should be "a very" not "an very".

" a vital rule" should be "a vital role"

I find the first paragraph of Section 4.3 quite abstruse and bare in its explanation of the method used.

**Experience Assessment:**

I have read many papers in this area.

**Review Assessment: Checking Correctness Of Derivations And Theory:**

N/A

**Review Assessment: Checking Correctness Of Experiments:**

I carefully checked the experiments.

**Review Assessment: Thoroughness In Paper Reading:**

I read the paper at least twice and used my best judgement in assessing the paper.

---

> ### Author Response · Authors · 2019-11-13
> **Response to reviewer 2 [3/3]**
>
> Q7: Would any other tasks from benchmarks like GLUE or SuperGLUE be amenable to your approach?
>
> A7: We believe generalizing our framework to other tasks is a promising direction to explore. We want to describe how to generalize our framework to more complex tasks, such as Question Answering.
>
> In this paper, we've made some attempts by applying NMET to a multi-hop multiple-choice Question Answering task (WikiHop in section 4.3) and it's making promising improvements. From the results, simply adding 100+ strictly-matched facts is not making a notable improvement, however, with the help of NMET, external knowledge from only 21 NL explanations and 5 rules help recognize a large number of structured facts and improves the performance.
>
> Extending our framework to span-based QA (e.g. SQuAD) is a research problem that we’re recently tackling, and we would like to show the following example.
>
> Question: How is packet switching characterized?
> Context: Circuit switching is characterized by a fee per unit of connection time, even when no data is transferred, while packet switching may be characterized by a fee per unit of information transmitted,
> Answer: by a fee per unit of information transmitted.
>
> A good explanation for such a dataset should cover three things: (1) What are the keywords in the question or context? (2) How is this type of question usually answered? (3) How do we locate the answer? For the above example, human annotators can write an NL explanation as follows: In the question, the words "packet switching" and "characterized" are important phrases. The question starts with "How is", so the answer probably starts with "by". The answer is right after "characterized" and the phrase "may be" is between "packet switching" and "characterized".
>
> We can easily adapt our CCG parser and conduct semantic parsing on the above explanation. It can be parsed into the following logical form (a little different from what we defined in this paper):
>
> - Q=Question, S=Sentence
> - Variables: X(NP), Y(VBZ), ANS(PP)
> - Rules1
> @Is(Q, @StartWith(“How is”))
> @Is(X, @Direct(@Right(“How is”)))
> @Is(Y, @Direct(@Right(X)))
> @Is(Q, @EndWith(Y))
> - Rules2
> @Is(X, @AppearIn(S))
> @Is(Y, @AppearIn(S))
> @Is(“may be”, @Between(X,Y))
> - Rules3
> @Is(ANS, @StartWith(“By”))
> - Rules4
> @Is(ANS, @Direct(@Right(Y))
>
> This logical form can then be applied to answer the following question:
>
> Question: How is cheese made?
> Context: Cheese is made the same way — by curdling milk — except the milk is curdled on purpose. Most cheese is made in factories. After milk is poured into big vats, a “starter culture” of bacteria is added to convert the lactose into lactic acid.
>
> To answer this question with the above rule, we first pre-process the question/context into chunks to fill the variables (X, Y) with candidate chunks in the question (e.g. X=cheese, Y=made). We score all candidates with Rule1. If the question matches with Rule1 with high confidence, we further retrieve relevant context with Rule2 in context. We later fill the variable ANS with selected sentence with Rule3. In the end we evaluate the candidate ANS with Rule4. The evaluation of candidates with Rule1-4 can be done with the proposed NMET framework. The answer to the above instance, "by curdling milk" can be correctly selected.
>
> Alternatively, we can leverage this LF to match context first (with Rule2) and then generate question (with Rule1) and answer (with Rule3-4).
>
> -------------------------------
> In addition, thanks for all the suggestions for improving the writing. We will correct all typos and grammatical errors.  Also, we’ll scrape the method summary and get straight into the Explanation Parsing section as suggested. We’ll explain LF(E) explicitly, which denotes applying logical forms generated by explanations directly onto instances.

---

> ### Author Response · Authors · 2019-11-13
> **Response to reviewer 2 [2/3]**
>
> Q4: "For keyword query q, we directly encode it into vector z_q by bi-LSTM and attention." Can you elaborate on how z_q is constructed?
>
> A4: We first use a BiLSTM to generate contextual embedding {h_i}, then all hidden layers are summarized by an attention layer as follows: [1]
>
> s_t = v_h^T \tanh (W_h h_t)
> a_t = \frac{\exp (s_t)}{\sum_{i=1}^n \exp (s_i)}
> c = \sum_{t=1}^n a_t h_t
>
> We’ll clarify this in detail in our paper.
>
> [1] Translation by Jointly Learning to Align and Translate   ICLR 2015
>
> -------------------------------
> Q5: I find the first paragraph of section 4.3 quite abstruse and bare in its explanation of the method used.
>
> A5: We apologize for not explaining it in detail. We’ve changed the first paragraph of section 4.3 as follows:
>
> To further test the capability of NMET in downstream tasks, we apply it to WikiHop country task by fusing NMET-matched facts into baseline model NLProlog. For a brief introduction, WikiHop country task requires a model to select the correct candidate ENT-Y for question ``Which country is ENT-X in?'' given a list of support sentences. As part of dataset design, the correct answer can only be found by reasoning over multiple support sentences.
>
> NLProlog is a model proposed for WikiHop. It first extracts triples from support sentences and treats the masked sentence as the relation between two entities. For example, ``Socrate was born in Athens'' is converted to (Socrate, ``ENT1 was born in ENT2'', Athens), where ``ENT1 was born in ENT2'' is later embedded by Sent2Vec to represent the relation between ENT1 and ENT2. Triples extracted from supporting sentences are fed into a Prolog reasoner which will do backward chaining and reasoning to arrive at the target statement country(ENT-X, ENT-Y). We refer readers to [1] for in-depth introduction of NLProlog.
>
> [1] NLProlog: Reasoning with Weak Unification for Question Answering in Natural Language, ACL 2019
>
> -------------------------------
> Q6: I am surprised that none of the modules or compared methods include any architecture that use a Transformer or a form of contextualized word vectors (McCann et al. 2017, Peters et al. 2017, Devlin et al. 2018). Is there an explanation for this?
>
> A6: The aim of our NMET framework is to leverage NL explanations so that models can learn to automatically annotate unlabeled data. The framework is model-agnostic as it can be integrated with any trainable base model (or downstream classifier). In our experiments, we choose BiLSTM+attention and ATAE-LSTM, which are generally used as base model for relation extraction and aspect level sentiment analysis respectively. And we demonstrate that NMET is significantly better than semi-supervised methods and data programming with these two base models.
>
> To address your concern, we also added BERT into Sentiment Analysis task (Restaurant) and the results are as follows:
>
> ——————————————————————————-
> Number of explanations   	    |        45     |       75        |
> Mean_Teacher (+BERT)           |      79.3    |      79.8      |
> Pseudo_Labeling (+BERT)	    |       78.7   |      81.0      |
> Self_Training (+BERT)              |       80.9   |      81.1      |
> NMET (+BERT)                          |       81.4   |      82.0      |
>
> The above results show that our model still outperforms baseline methods when BERT is incorporated. The gain is not as remarkable as in our paper, because the performance is approaching the upper bound (around 85, we get this score by feeding all data to BERT and conduct supervised training). Results of NMET is close to this upper bound, with only 75 explanations, which again demonstrates the annotation efficiency with NMET. The results align with our conclusions and we’ll add it in our paper.

---

> ### Author Response · Authors · 2019-11-13
> **Response to reviewer 2 [1/3]**
>
> Thank you for your suggestions for improving the presentation of our work! Our response is as follows:
>
> -------------------------------
> Q1: One recent work that comes to mind from ACL 2019: Leveraging Language Models for Commonsense Reasoning (Rajani et al. 2019). And it seems like a reasonable reference to contrast the more structured approach to using explanations like Srivasta et al. (2017), Hancock et al. (2018), and this work.
>
> A1: Thank you for pointing out this line of work and we’ll add it into reference. We would like to emphasize that both methods (language model and structured approach) leverage explanations but with different focus and motivation.
>
> Rajani et al. (2019) explore an implicit and data-hungry method (language models) to leverage natural language explanations. They focus on the importance of NL explanations as *supplementary supervision*, when *labeled data is abundant*.
>
> Our work (also [1] and [2]) proposes a more structured and explicit approach. We focus on leveraging NL explanations to *efficiently annotate unlabeled data*, when *annotation is expensive*. Our setting is most similar to Stanford Snorkel [3], where labeling training data is the largest bottleneck (especially when domain expertise is required). In these settings, users write labeling functions (in our case, it is NL explanations) that express arbitrary heuristics to create training data rapidly and efficiently.
>
> [1] Training Classifiers with Natural Language Explanations   ACL 2018
> [2] Joint concept learning and semantic parsing from natural language explanations   EMNLP 2017
> [3] Snorkel: Rapid training data creation with weak supervision   VLDB Endowment 2017
>
> -------------------------------
> Q2: Why is it so essential that you study the setting in which explanations are low-resources? I'm curious to see what would happen with more explanations.
>
> A2: As is stated in A1, our setting is most similar to *Snorkel*. We focus on how to leverage NL explanations *efficiently* when annotation is expensive, and we demonstrate *efficiency* by showing promising results with only *a few* NL explanations in our experiments. This setting is very realistic in certain domains that require expertise (e.g. medical domain), where large-scale conventional annotation is not practical due to time and budget limit.
>
> Our experiments in Sec 4.2 demonstrate our model benefits from more explanations (as shown in Fig. 5 on page 9 or the table below). In domains where annotation is not expensive, we can certainly collect more explanations for improvement. But this kind of setting is not the focus of our study. We agree that there are interesting topics like if there is a “saturation point” for using explanations as supervision and will leave them as future work.
>
> Sentiment Analysis (Restaurant):
> ——————————————————————————-—-——-—
> Number of explanations   |  10   |  20   |  30   |  45   |  60   |  75   |
> Pseudo Labeling                 | 61.0 | 71.2 | 70.3 | 70.9 | 70.8 | 70.7 |
> LSTM+ATT                            | 64.7 | 71.2 | 70.8 | 71.1 | 71.2 | 70.8 |
> MeanTeacher                      | 51.2 | 70.5 | 72.2 | 72.0 | 72.2 | 76.0 |
> NMET                                    | 72.1 | 72.3 | 74.6 | 75.8 | 76.2 | 76.9 |
>
> Relation Extraction (TACRED):
> ———————————————————————————-——-—
> Number of explanations   | 100  | 120  | 140  | 170  | 200  | 230  |
> Pseudo Labeling                 | 37.5 | 37.1 | 40.6 | 41.5 | 42.1 | 42.1 |
> ATAE-LSTM                           | 36.8 | 37.8 | 40.4 | 41.4 | 41.0 | 41.9 |
> MeanTeacher                      | 36.5 | 36.9 | 40.9 | 40.8 | 41.7 | 42.5 |
> NMET                                    | 40.9 | 42.1 | 43.0 | 45.6 | 45.1 | 45.8 |
> ——————————————————————————-—-——-—
>
> -------------------------------
> Q3: Can you elaborate on why LF(E) is so dominant on precision in Table 6 and 7 of the Appendix, but low on recall, rather than just saying this is expected in section 4.1?
>
> A3: As is found in our collected dataset, people tend to use detailed and specific constraints in an NL explanation to ensure they cover all aspects of the instance. As a result, those instances that satisfy the constraints are correctly labeled in most cases, and thus the precision is high.
>
> Meanwhile, generalization ability is compromised. As discussed in the introduction, linguistic variants are common, which makes it difficult to generalize NL explanations for matching sentences that are semantically equivalent but having different word usage, e.g. “is founded by” in a rule can not be strictly matched with “is created by” though they express the same meaning. In this case, those “is created by” instances cannot be matched, and thus the recall is low. We will clarify this in detail in our paper.

---

### Author Response · Authors · 2019-11-13
**Paper Revised**

We would like to thank all the reviewers for their efforts and valuable comments. We have revised the paper to address the questions from reviewers. The major updates are:

- Improved clarity in section 3.1 and 4.3, in response to reviewer #2 and #3.
- Scrape the method summary and get straight into the Explanation Parsing section in response to reviewer #2.
- Correct all typos and grammatical errors and add one related work in response to reviewer #2.

---

### Decision · Program_Chairs · 2019-12-19

**Decision:**

Accept (Poster)

**Comment:**

This paper proposing a framework for augmenting classification systems with explanations was very well received by two reviewers, and on reviewer labeling themselves as "perfectly neutral". I see no reason not to recommend acceptance.